# Altered feeding behavior and immune competence in paper wasps: A case of parasite manipulation?

Laura Beani[1]*, Marta Mariotti Lippi[1], Nadia Mulinacci[2], Fabio Manfredini[3,4], Lorenzo Cecchi[2], Claudia Giuliani[5], Corrado Tani[2], Niccolò Meriggi[1], Duccio Cavalieri[1]*, Federico Cappa[1]

1 Dipartimento di Biologia, Università di Firenze, Firenze, Italia, 2 Dipartimento di NEUROFARBA, Università di Firenze, Firenze, Italia, 3 School of Biological Sciences, Royal Holloway University of London, Egham, United Kingdom, 4 School of Biological Sciences, University of Aberdeen, Aberdeen, United Kingdom, 5 Dipartimento di Scienze Farmaceutiche, Università degli Studi di Milano, Milano, Italia

* laura.beani@unifi.it (LB); duccio.cavalieri@unifi.it (DC)

**Data Availability Statement:** Data are available at https://osf.io/jfvm2/quickfiles.

## Abstract

Paper wasps (*Polistes dominula*), parasitized by the strepsipteran *Xenos vesparum*, are castrated and desert the colony to gather on plants where the parasite mates and releases primary larvae, thus completing its lifecycle. One of these plants is the trumpet creeper *Campsis radicans*: in a previous study the majority of all wasps collected from this plant were parasitized and focused their foraging activity on *C. radicans* buds. The unexpected prevalence and unusual feeding strategy prompted us to investigate the influence of this plant on wasp behavior and physiology through a multidisciplinary approach. First, in a series of laboratory bioassays, we observed that parasitized wasps spent more time than non-parasitized ones on fresh *C. radicans* buds, rich of extra-floral nectaries (EFNs), while the same wasps ignored treated buds that lacked nectar drops. Then, we described the structure and ultra-structure of EFNs secreting cells, compatible with the synthesis of phenolic compounds. Subsequently, we analysed extracts from different bud tissues by HPLC-DAD-MS and found that verbascoside was the most abundant bioactive molecule in those tissues rich in EFNs. Finally, we tested the immune-stimulant properties of verbascoside, as the biochemical nature of this compound indicates it might function as an antibacterial and antioxidant. We measured bacterial clearance in wasps, as a proxy for overall immune competence, and observed that it was enhanced after administration of verbascoside—even more so if the wasp was parasitized. We hypothesize that the parasite manipulates wasp behavior to preferentially feed on *C. radicans* EFNs, since the bioactive properties of verbascoside likely increase host survival and thus the parasite own fitness.

## Introduction

A parasite can deeply change the life-history of its host. If parasite-induced changes improve the transmission of the parasite, they can be considered as adaptive manipulation rather than a mere by-product of pathology [1–4]. Different altered behaviors have been described in the

**Funding:** Financial support to LB was provided by the University of Florence.

**Competing interests:** The authors have declared that no competing interests exist.

extended phenotype paradigm [5, 6], object of a lively debate [7]. The outcome of the manipulative efforts is not necessarily a novel and bizarre behavior, but can rather be the altered expression of usual behavioral patterns of the host [8], for example subtle changes in feeding preference [9] or the modulation of core biological functions such as nervous and immune systems [10–12]. One example of a complex manipulation is the interaction between the endoparasitic insect *Xenos vesparum* (Rossi)(Strepsiptera, Xenidae) [13, 14] and its host, the primitively eusocial wasp *Polistes dominula* (Christ)(Hymenoptera, Vespidae). If parasitized, putative workers are castrated, do not participate in colony tasks, and desert the nest to form aggregations on nearby plants, where *X. vesparum* completes its life cycle by mating and releasing infective larvae [15–18]. The behavior of parasitized wasps has some features that remind the non-reproductive phase of future queens: they are in ovary diapause, do not perform any colony task and overwinter in sheltered aggregations [19]. As a matter of fact, parasitized wasps are often found together with future queens in overwintering aggregations and in a previous study Geffre et al. [20] reported that the parasite manipulates the behaviour of the host to make it more queen-like by specifically shifting the expression of caste-related genes [20].

In a long-term study, Beani et al. [17] described the strong attraction of parasitized wasps towards *Campsis radicans* (L.)(Bureau)(Bignoniaceae), a perennial plant commonly known as "trumpet creeper". This plant is highly attractive for pollinators both in its North America native range and in Europe, where it has been introduced [21, 22]. Two thirds of all wasps that were sampled on trumpet creepers were parasitized, whereas the parasite prevalence was much lower (only 6%) in the broader study area. Parasitized wasps aggregated on *C. radicans* and fed on the calyx of buds, touching the sepals with antennae and scraping their surface with mandibles (Fig 1, S1 Video). Long patrols and aggressive response towards any approaching wasp or

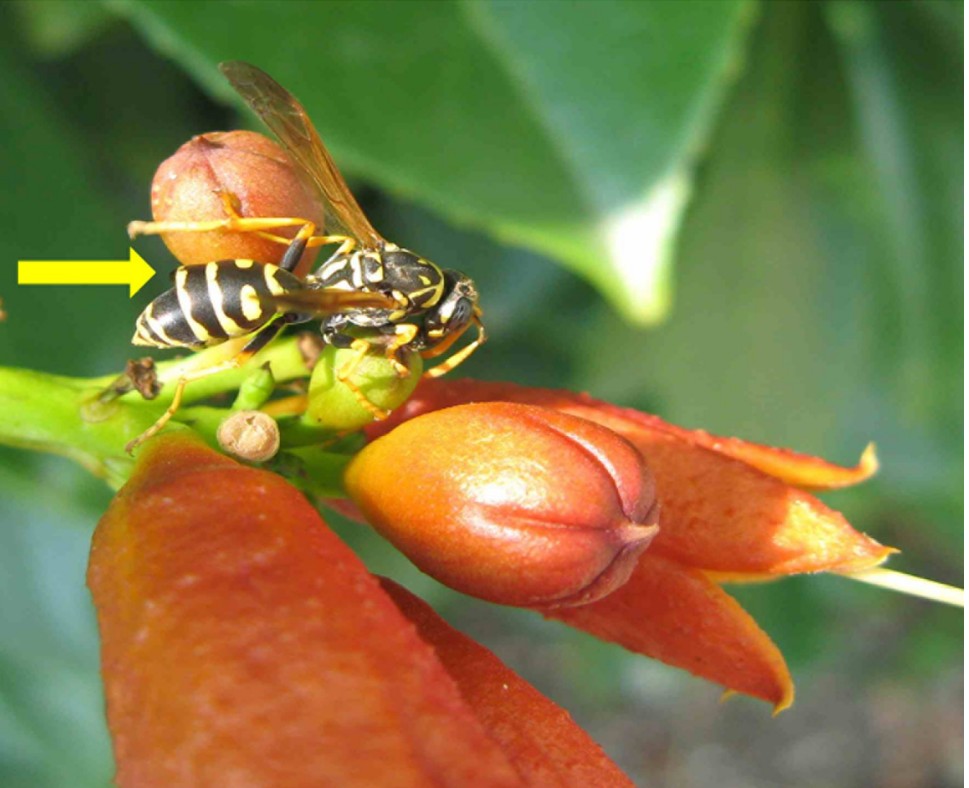

**Fig 1. *Parasitized P. dominula on C. radicans.*** A *P. dominula* wasp, parasitized by one male *X. vesparum* (arrow: See the tergite lifted by the parasite's *puparium*), is feeding on *C. radicans* buds.

other insect were more frequent on buds than on leaves and flowers [17]. The unusual high number of extrafloral nectaries (EFNs) on calyx, corolla, petiole and fruit has been interpreted as an advanced strategy of *C. radicans* to attract ants as bodyguards for defence against herbivore insects [23, 24]. Interestingly, previous research has shown that nectar secondary metabolites, produced by EFNs, are able to manipulate the behavior of ants [25]. However, very little is known on feeding behavior focused on EFNs in other insects, like wasps [26, 27].

In this study, we characterize the complex interaction that links wasps, parasites and trumpet creepers using a step-by-step empirical and multidisciplinary approach, that is often missing in studies on parasite manipulation [7–28]. We first performed a series of laboratory bioassays to evaluate the feeding preference of wasps (in the presence or absence of the parasite) towards fresh buds, rich of secreting EFNs, or treated buds, deprived of these secretions. Second, to assess whether peculiar traits of the EFNs may explain wasps' attraction towards trumpet creepers, we explored the distribution, the structure and the ultrastructure of EFNs secreting cells covering the calyx of fresh buds by means of optic and scanning electron microscopy and histochemical analysis. Then, using HPLC-DAD-MS, we analysed the content of the secondary metabolites obtained from the extracts of *C. radicans* buds at different stages of growth, with and without secreting EFNs. Finally, we tested whether the administration of verbascoside, the most abundant phenolic compound found in *C. radicans* extracts of sepals rich of EFNs, can act as an immunostimulant in parasitized and non-parasitized wasps: this investigation was motivated by the fact that verbascoside is known for its bioactive properties. We hypothesize a scenario of parasite manipulation, where the selective feeding behavior of parasitized wasps, focused on EFNs, activates their immune system through the intake of verbascoside.

## Materials and methods

### Sample collection and rearing

In June 2018 we collected 25 *P. dominula* females parasitized by *X. vesparum* (13 infected by one male, 12 by one female) and 20 non-parasitized wasps, from 3 flowering bushes of *C. radicans* in the plain of Sesto Fiorentino (Florence, Italy: 43° 50' 7" N, 11° 11' 46" E). A second set of wasps emerged in captivity from 16 colonies collected in the same area: these were 24 parasitized wasps (13 infected by one male, 11 by one female) and 24 non-parasitized wasps. Wasps (93 in all) were housed in glass boxes (15x15x15 cm) at room temperature (26 ± 2 C°), with water and sugar ad libitum. The head-width of wasps was measured with a dial caliper to the nearest 0.05 mm as proxy for body size [29]. This was used to unveil any possible effect of parasitism on host size that could affect feeding behavior.

### Preference bioassays

To investigate the attractive features of *C. radicans* buds, *P. dominula* wasps were subjected to feeding preference tests in relation to parasite (parasitized or non-parasitized), rearing environment (emerged in the field or in the laboratory), bud's (fresh or treated) and different bud's treatments. We presented wasps with 3 binary choices: an untreated fresh bud, collected just before the test, versus a treated bud: (trial 1) a withered bud, detached from the plant 24 h before; (trial 2) a fresh bud washed for 5 min in a mixture of ethanol:water 1:1 or (trial 3) hexane:water 1:1, in order to remove potentially attractive polar and apolar components. The experimental apparatus consisted in a glass box (15x15x15 cm) with two circular holes at the opposite corners of the top panel, where two 3 cm long buds protruded from 1.5 ml Eppendorf tubes for half their size. Following each trial, tubes and buds were replaced and the apparatus was carefully washed with 96% ethanol.

A single wasp was introduced in the apparatus and tested only once. Trials were performed on three different days between 11.00 and 16.00 hours, when wasps are more active. Sugar was removed from the home cage 24 h before the trial, in order to promote foraging activity. Ten out of 93 wasps (5 parasitized and 5 non-parasitized) were inactive and did not approach any bud; these subjects were therefore excluded from the analysis. One observer, blind to the experimental conditions, recorded directly for how long the wasp rubbed the tips of its antennae (i.e. "antennation", to detect chemical cues from the substrate) and scratched the surface of the buds with its jaws, moving rhythmically its abdomen during ingestion (i.e. "feeding"). The duration of antennation and sucking were clumped together as "time spent on buds"; we used these temporal data as usual proxy for wasp's choice index [30]. Since the time spent on buds did not last more than 30 sec, we opted to measure 0–30 sec from the first contact to each stimulus bud. Our bioassays were designed to simulate a "natural foraging" scenario where a wasp quickly assesses different buds and chooses the preferred one (S1 Video).

## Structure and ultrastructure of Extra-Floral Nectaries (EFNs)

Since the feeding strategy of parasitized wasps is focused on EFNs of the calyx, we analysed the distribution, activity and morphology of EFNs under a Leitz DM-RB Fluo Optic microscope equipped with a digital Nikon DS-L1 camera (LM) and a QUANTA-200 FEI environmental scanning electron microscope (ESEM), operating at low vacuum (pressure 0.53 Torr).

To describe the synthesis activity of EFNs, histochemical features of their secretions were observed under a light microscope (LM). Samples were fixed in FAA solution for 7 days, embedded in historesin (Technovit 7100; Radnor, PA, USA) and dyed with Toluidine Blue for a general survey, Sudan III/IV for total lipids, PAS reaction for total polysaccharides, and Ferric trichloride for polyphenols. Moreover, ultrastructure was described under a Philips EM-300 transmission electron microscope (TEM). Calyx fragments were fixed in 2.5% glutaraldehyde and 0.1 M phosphate buffer at pH 7.2, post-fixed in 2.0% osmium tetroxide, dehydrated in ascending ethanol series up to absolute and embedded in Spurr resin. The ultrathin sections were contrasted with uranyl acetate and lead citrate.

## Chromatographic analysis of *C. radicans* extracts

To characterize the chemical profile of EFNs, in 2016 and 2017 we extracted 200 mg of fresh tissue of calyx (sepals with and without developed EFNs) for 24 hours with 2.5 ml ethanol or hexane. Small and large buds (0.5–1 and 2–3 cm long, respectively), at different stages of growth, were treated as for calyx to evaluate the concentration of secondary metabolites in EFNs. The solutions obtained after centrifugation were analysed by HPLC-DAD-MS (High-Performance Liquid Chromatography—Diode-Array Detector- Mass).

Ethanol and hexane extracts were analyzed with an HP 1100L Liquid Chromatograph, provided with a Diode Array Detector (DAD) and coupled with a TOF Mass Spectrometer equipped with an electrospray (ESI) interface (all from Agilent Technologies, Palo Alto, CA, USA). The analysis was performed in negative ion mode with drying gas ($N_2$) temperature 350˚C; drying gas flow-rate 6 l/min; nebulizer 20 psi; capillary voltage 4000 V; fragmentation 150 V; skimmer 60 V. A 150 mm × 3.0 mm i.d., 3.5 μm Eclypse plus C18 column (Agilent, USA) was employed. The eluents for the mobile phase were water (pH 3.2 formic acid) and $CH_3CN$, applying a multi-step linear solvent gradient: 0–20 min 5–65% B; 20–22 min 65–98% B; 22–24 min 98% B; 24–25 min 98–5% B; flow rate was 0.4 ml min$^{-1}$. Quantitative analysis was carried out using a six-point calibration curve at 330 nm built and pure verbascoside as external standard, with linearity range of 0–1.96 μg and $R^2$ 0.9996.

## Administration of verbascoside and immunocompetence assay

Given the abundance of verbascoside in the extracts of fresh buds rich of EFNs (see Results, *Chromatographic analysis of C.* radicans), we evaluated whether the intake of verbascoside could have an effect on the immunocompetence of wasps. In June 2018, 75 *P. dominula* wasps (of which 25 parasitized by one female *X. vesparum*) were collected in the plain of Sesto Fiorentino (Florence, Italy) and housed as in the bioassay experiment. 2 μl of the solution of sugar or maltodextrin or verbascoside, diluted in sterile water to the concentration of 125 mg/ml, was administrated to each wasp with a micropipette every other day for three times, thus avoiding a high dose in a single feeding event.

Wasps were assigned to 5 treatment sub-groups: 17 non-parasitized wasps were fed with 10% sugar solution (control group); 5 non-parasitized wasps were fed with 10% maltodextrin solution (maltodextrin group, second control group); 10 parasitized wasps were fed with 10% sugar solution (parasite group); 28 non-parasitized wasps were fed with diluted extract of verbascoside (verbascoside group); 15 parasitized wasps were fed with diluted extract of verbascoside (verbascoside+parasite group). Since one drop of the solution (2 μl) contained 0.250 mg of verbascoside, the total amount of verbascoside assumed by a single wasp was 0.750 mg, approximately corresponding to the verbascoside content per bud (see Table 1).

The immune challenge was performed 7 days after the first administration. As a measure of general immune competence, we used a standard approach for insects that consists in quantifying the ability of a specimen to remove bacterial cells from the haemolymph [31, 32]. Wasps were infected through injection with the tetracycline-resistant strain XL1 Blue *Escherichia coli* (Stratagene, La Jolla, California), an immune elicitor not found in *P. dominula* and commonly used to test antibacterial activity in insects [33–39].

Bacterial cultures were grown overnight aerobically in Luria-Bertani (LB medium: 1% bacto-tryptone, 0.5% yeast extract, 1% NaCl) complex medium containing tetracycline at a concentration of 10 $\mu$g ml$^{-1}$ at 37 °C in a shaking incubator. After centrifugation, bacteria were washed twice in phosphate-buffered saline (PBS), resuspended and diluted to the desired concentration with PBS ($\sim 1,5 \times 10^8$ cells ml$^{-1}$). The estimated amount of bacterial cells in the solution was measured as optical density with an Eppendorf BioPhotometer® 6131. Each wasp was then infected through injection of 1 $\mu$l of inoculum, containing $\sim 1,5 \times 10^5$ cells, with a HamiltonTM micro syringe between the second and third tergites [39]. After injection, wasps were maintained in the dark at room temperature (20°C) for 24h and then dissected to

**Table 1. ANOVA table (type II tests).**

| Factor | Sum Sq | Df | F value | Pr (>F) |
|---|---|---|---|---|
| **Parasite** | 502.5 | 1 | 32.582 | 5.602e-08 *** |
| **Rearing environment** | 61.2 | 1 | 3.971 | 0.048* |
| **Bud condition** | 7000.5 | 1 | 453.899 | 2.2e-16 *** |
| **Bud treatment** | 133.3 | 2 | 4.323 | 0.015 * |
| **Parasite: Rearing** | 1.7 | 1 | 0.111 | 0.738 |
| **Parasite: Bud condition** | 607.8 | 1 | 39.408 | 3.253e-09 *** |
| **Parasite: Bud treatment** | 9.0 | 2 | 0.290 | 0.748 |

Factors influencing feeding preference (i.e. time spent on buds) estimated in the analysis were reported as follows: Parasite (parasitized and non-parasitized wasps), Rearing environment (wasps emerged in the field and in the laboratory), Bud condition (fresh bud and treated bud) and Bud treatment (trial 1: withered; trial 2: hexane-washed; trial 3: ethanol-washed). Tested interaction effects were also reported (Parasite:Rearing, Parasite:Bud condition, Parasite:Bud treatment). Significance codes: p<0.05*, p<0.01**, p<0.001***.

remove the parasite, the sting and the venom sac, in order to avoid any possible interference with bacterial counts of the parasite or the antimicrobial peptides in the venom [40]. Wasps were homogenized in 1ml of PBS solution using sterile pestle. The homogenate was serially diluted and plated on LB solid medium plus tetracycline 10μg/ml, then incubated at 37˚C overnight. The following day bacterial clearance was evaluated by count of bacterial Colony Forming Units (CFUs)/ml per wasp.

## Statistical analysis

Bioassay and bacterial clearance analyses were carried out into R environment v4.0.2 [41]. Feeding preference was measured as time (sec) spent by wasps on each bud. The preference bioassays analysis was assessed by using linear model with factorial design fitted with *lm*() function [42, 43], then type II analysis of variance (ANOVA) from *car* package was performed [44]. Factors influencing feeding preference estimated in the analysis were: (i) parasite (parasitized and non-parasitized wasps); (ii) rearing environment (wasps emerged in the field and in the laboratory); (iii) bud condition (fresh bud and withered/treated bud); and (iv) bud treatment. The linear model included the interaction term which is added in in the formula. The comparisons of least-square means for significant interaction were performed using *lsmeans*() function from *lsmeans* package [45].

   Regarding the sex of parasite, differences in time spent on buds by wasps with male or female parasite were preliminarily assessed using Kruskal-Wallis rank sum test. Correlation between wasps size and time spent on bud was tested using Pearson correlation test (confidence level = 0.95) through *cor.test*() function.

   Concerning the bacterial clearance assay, differences in CFU/ml levels among treatment groups were tested using Kruskal-Wallis rank sum test through *kruskal.test*() function. Multiple comparisons following a significant Kruskal-Wallis test were performed using Dunn's Kruskal-Wallis Multiple Comparisons test thorugh *dunnTest*() function (Benjamini-Hochberg method) from *FSA* package [46]. Boxplots were generated using *ggplot2* package [47].

## Ethics statement

The authors certify that procedures involving experimental animals were performed in compliance with Italian animal welfare laws, guidelines and policies.

## Results

### Preference bioassays

In 3 feeding preference trials, all the wasps briefly (mean time: 5.01 ± 0.21 sec) approached the withered, ethanol- and hexane-washed buds, while they foraged (up to 30 sec) only on untreated fresh buds. Fresh buds were 4 times more attractive than treated buds (S1 Table, in S1 File). Under optic microscope we verified that all the 3 treatments had removed the shiny nectar drops, secreted by EFNs throughout blooming [23]. Parasitized wasps spent significantly more time (p<0.001) on fresh buds than the non-parasitized wasps (Fig 2, Table 1). However, the bioassay test also showed that the parasitized condition was not the only factor influencing the wasps behavior, indeed the other factors tested in the model were also significant, suggesting that the time spent on buds was also dependent on the bud condition, rearing environment and bud treatments (Table 1, Fig 2). Although the main factors were all significant, only the interaction between parasitized condition and bud condition was statistically significant (p<0.001).

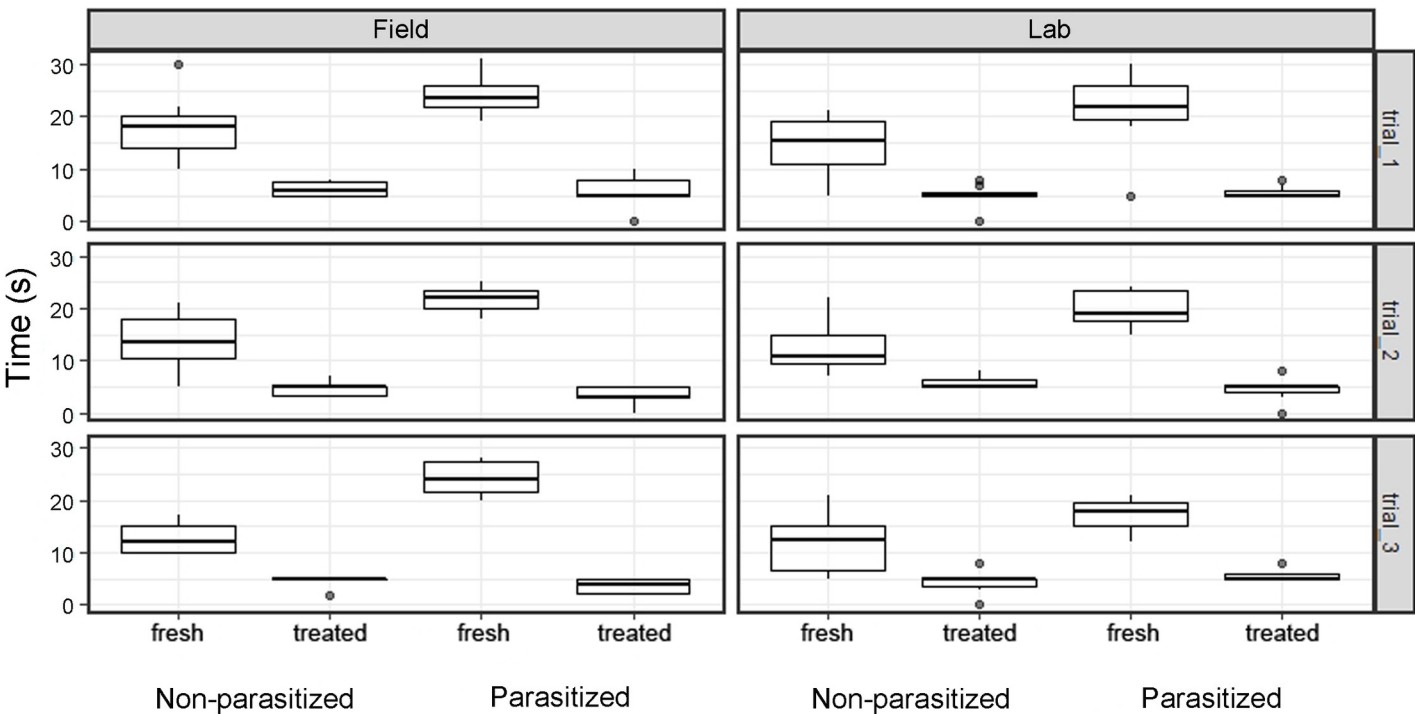

**Fig 2. Preference bioassay.** Box plots show the time (0–30 sec) spent by wasps on buds in relation to bud condition (fresh vs treated), parasite (parasitized vs non-parasitized) and rearing environment (field vs lab). Field: N = 20 non-parasitised wasps, N = 25 parasitized; lab: N = 24 non-parasitised wasps, N = 24 parasitized. The plots were displayed considering the bud treatment (trial 1: Withered; trial 2: Hexane-washed; trial 3: Ethanol-washed).

The synergistic effect produced by the interaction showed an increase in the time spent by parasitic wasps on fresh buds (interaction plot, S1 Fig in S1 File). Treated buds were always avoided, regardless of parasite occurrence (Fig 2; S2 Table, S1 Fig in S1 File).

There was a weak significance in wasps emerged in the field rather than in the laboratory to be more attracted towards fresh buds (Fig 2, F = 3.97, P = 0.048), probably due to their previous foraging experience on trumpet creepers. Overall, chemical cues influenced wasps' choice more than chromatic cues, since the orange colour of buds was retained after the different bud treatments. There was no significant difference in head width among the 4 experimental groups (ANOVA test: df = 3, F = 0.22, p = 0.87), neither between wasps emerged in the field and in the laboratory (df = 1, F = 0.079, p = 0.81). Time spent on the bud was not related to body size (Pearson correlation test; t = -0.63, df = 164, p = 0.53). Parasite sex did not influence the time spent on untreated fresh buds in our sample of 44 wasps, 22 infected by one male and 22 infected by one female *X. vesparum*, when the 3 bioassays are analysed together (Kruskal-Wallis test: Chi-sq = 0.18, df = 1, p = 0.67).

### Structure and ultrastructure of calyx extra-floral nectaries (EFNs)

The nectar drops secreted by EFNs were clearly visible only on fresh buds before anthesis (LM, Fig 3A). Isolated or grouped EFNs (from 0 to 9 per sepal) were irregularly scattered on the abaxial surface of the calyx teeth, the tips of sepals (Fig 3A–3C), thus, we could select tissues with and without EFNs in secreting stage for chemical analysis. They showed a circular shape, with a central large depression where the nectar drop was secreted. EFNs were located in shallow epidermal concavities, so that their peripheral rim was hardly higher than the calyx

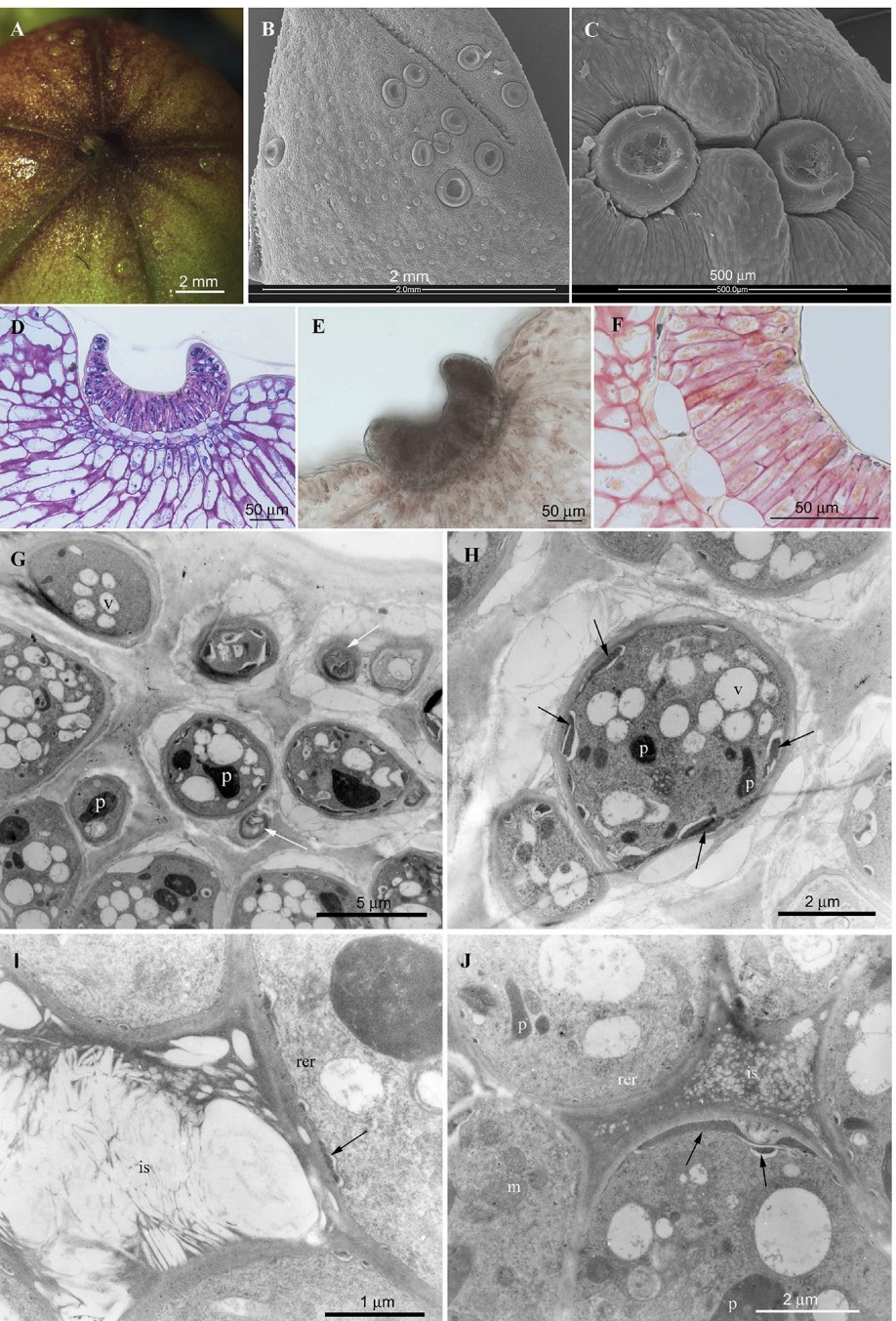

**Fig 3.** *Campsis radicans* **extrafloral nectaries (EFNs).** LM: A) a fresh large bud with evident EFNs irregularly distributed on the tips of the sepals. ESEM: B) isolated and grouped EFNs with circular to oval shape on the teeth of the calix; C) two circular EFNs in an epidermal concavity. LM: D-F) medial longitudinal section of EFN: (D) Toluidine Blue E) FeCl₃ F) PAS. TEM: G) secreting cells at the EFN apical portion with signals of apoptosis, *i.e.* organelle degradation and myelin-like figures (arrows); H) secreting cell with evident periplasmic spaces (arrows); I) intercellular space containing osmiophilic multi-laminar material; J) secreting cells and intercellular spaces at the EFN basal portion with osmiophilic globular inclusions (arrows) in the periplasmic spaces and in the cell wall matrix. Symbols: *is, intercellular space; m, mitochondrion; p, plastid; rer, rough endoplasmic reticulum; v, vacuole.*

surface. LM observations showed that EFNs were formed by a layer of columnar secreting cells, closely tightened to each other (Fig 3D and 3E), standing on a layer of short, squared cells ("cuboidal cells", according [24]). The cytoplasm of the secreting cells was well stained by Toluidine Blue (Fig 3D), Sudan III-IV and FeCl₃ (Fig 3E), while it was slightly stained by PAS (Fig 3F).

TEM observations revealed that the dense cytoplasm of the secreting cells contained variable-sized vacuoles, mitochondria, globular or ellipsoid plastids, with osmiophilic homogeneous stroma, well-developed rough endoplasmic reticulum (*rer*) with dilated vesicles (Fig 3G–3J). Some cells had evident apoptosis marks (Fig 3G and 3H). Osmiophilic deposits, indicating the production of lipidic substances, and osmiophilic globular inclusions were observed in the periplasmic spaces and in the cell wall matrix; the intercellular spaces were filled with material of heterogeneous appearance (Fig 3H and 3I). A large vacuole almost entirely occupied the cuboidal cells, presumably involved in the secretion process. In conclusion, the histochemical features of EFNs covering *C. radicans* buds, the extensive development of plastids and *rer* profiles were all compatible with the synthesis and transfer of phenolic compounds and indicate the production of lipidic substances which pass through the wall.

## Chromatographic analysis of *C. radicans*

All the hexanoic extracts showed no analytes in the chromatographic profiles, thus we focused on the ethanol extracts. Due to the irregular distribution of EFNs on sepals (Fig 3A), we compared tissues with and without EFNs in secreting stage. The chromatographic profile at 330 nm revealed only one main peak in the ethanolic extracts from tissues rich of EFNs (Fig 4), which was identified as verbascoside according to retention time, UV and mass spectra when compared to the verbascoside standard (S2 and S3 Figs in S2 File). Verbascoside, the most abundant secondary metabolite in tissues rich in EFNs, is a water-soluble phenolic compound belonging to the phenylethanoid glycoside family. We prepared a dry extract of verbascoside, titrated at 10% w/w, with the remaining part constituted by traces of verbascoside isoforms and maltodextrins, oligosaccharides used as a food additive. Usually the ethanol extracts of herbal products are characterized by the co-presence of several compounds belonging to different chemical classes. Unexpectedly, the ethanol extracts of tissues rich of EFNs showed that verbascoside was largely predominating, with the other metabolites only present in negligible amount. This result was confirmed by the samples collected in two successive years and by the chromatograms at 280 and 330.

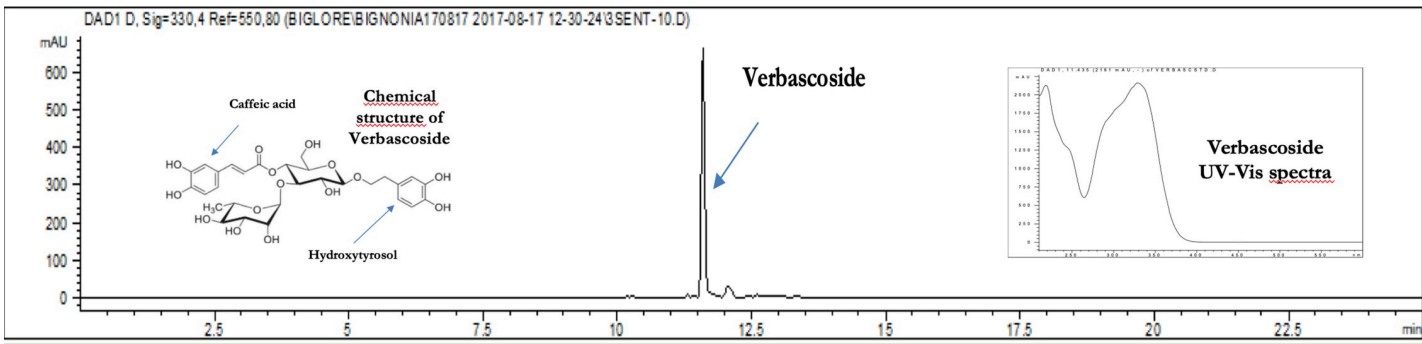

**Fig 4. Chromatographic profile of the ethanolic extracts from *C. radicans* tissues rich of EFNs.** The chemical structure of Verbascoside is characterized by two phenolic moieties (a caffeic acid and a hydroxytyrosol) attached to the positions 1 and 4 of a β-glucopyranose belonging to the α-rhamnopyranosyl-β-glucopyranose disaccharide moiety. Chromatographic profile at 330 nm of sepals with EFNs and its UV-Vis spectra.

**Table 2. Content of verbascoside expressed as mg of verbascoside per gram of tissue and as mg of verbascoside per bud.**

| Sample | mg/g | mg/bud |
|---|---|---|
| Calyx with secretory EFNs | 7.45 | - |
| Calyx without secretory EFNs | 4.03 | - |
| Small buds | 2.78 | 0.79 |
| Large buds | 1.13 | 0.76 |

With regards to the amount of verbascoside detected, the tissue with fully developed EFNs had the highest amount, approximately 7.5 mg/g, almost double than the samples not covered by EFNs (Table 2), suggesting that EFNs in secreting stage contain the highest concentration of verbascoside. When the concentration was expressed in mg of fresh tissue, small buds (0.5–1 cm long) showed a content of verbascoside per bud that was more than double if compared with that of large buds (2–3 cm long), due to the number of EFNs in secreting stage, higher in 7 small buds than in 3 large buds. The function of elevated concentration of verbascoside in younger buds was likely the defense from florivores. However, when the content of verbascoside was expressed as mg per bud, the difference was minimal: 0.76 mg$_{verb}$ for large buds and 0.79 mg$_{verb}$ for small buds.

### Immunocompetence assays

Wasps treated with extracts of verbascoside, both parasitized and non-parasitized, showed a significant reduction of *E. coli* loads compared to wasps fed with sugar or maltodextrin solutions (Fig 5). After Kruskal-Wallis rank sum test (Chi-sq = 43.524, df = 4, p < 0.001) a post-hoc analyses among groups through Dunn's test (Table 3) revealed that controls differed from all other groups, except for the maltodextrin group. Parasitized wasps treated with verbascoside (verbascoside + parasite) displayed a further reduction of bacterial loads, significantly lower than all other groups (Table 3). No significant differences in bacterial clearance were observed between parasite and verbascoside groups.

### Discussion

In this study we report that *P. dominula* wasps, if parasitized by *X. vesparum*, focused their foraging activity on extra-floral nectaries (EFNs) which cover *C. radicans* buds. We investigated this unusual feeding behavior by means of a combined approach of structural and chemical analyses, and preference and immune assays. We present four main experimental results: (1) wasps show a marked preference for fresh untreated buds of *C. radicans*, covered with EFNs, than for treated buds, deprived of nectar drops and, when parasitized, they spend more time foraging on fresh buds than non-parasitized wasps; (2) fresh buds of trumpet creepers are rich in EFNs, whose histochemical and structural features denote the secretion of phenolic compounds; (3) the most abundant compound in these secretions is verbascoside, a bio-active molecule known in traditional medicine for its antioxidant, anti-inflammatory, antibacterial, antineoplastic and antifungal properties [48–50]; (4) ingestion of verbascoside enhances wasps' ability to clear a bacterial infection, especially when parasitized.

The marked attraction of parasitized wasps to *C. radicans* flowering bushes, here confirmed by preference assays, is intriguing. Wasps preferred to forage on fresh untreated buds, regardless of parasite or treatment. Thus, visual cues alone, i.e. the bright orange colour of buds, retained also by buds withered or washed with solvents, were not the main driver for wasps' choice compared to chemical cues. Interestingly, parasitized wasps spent more time than non-

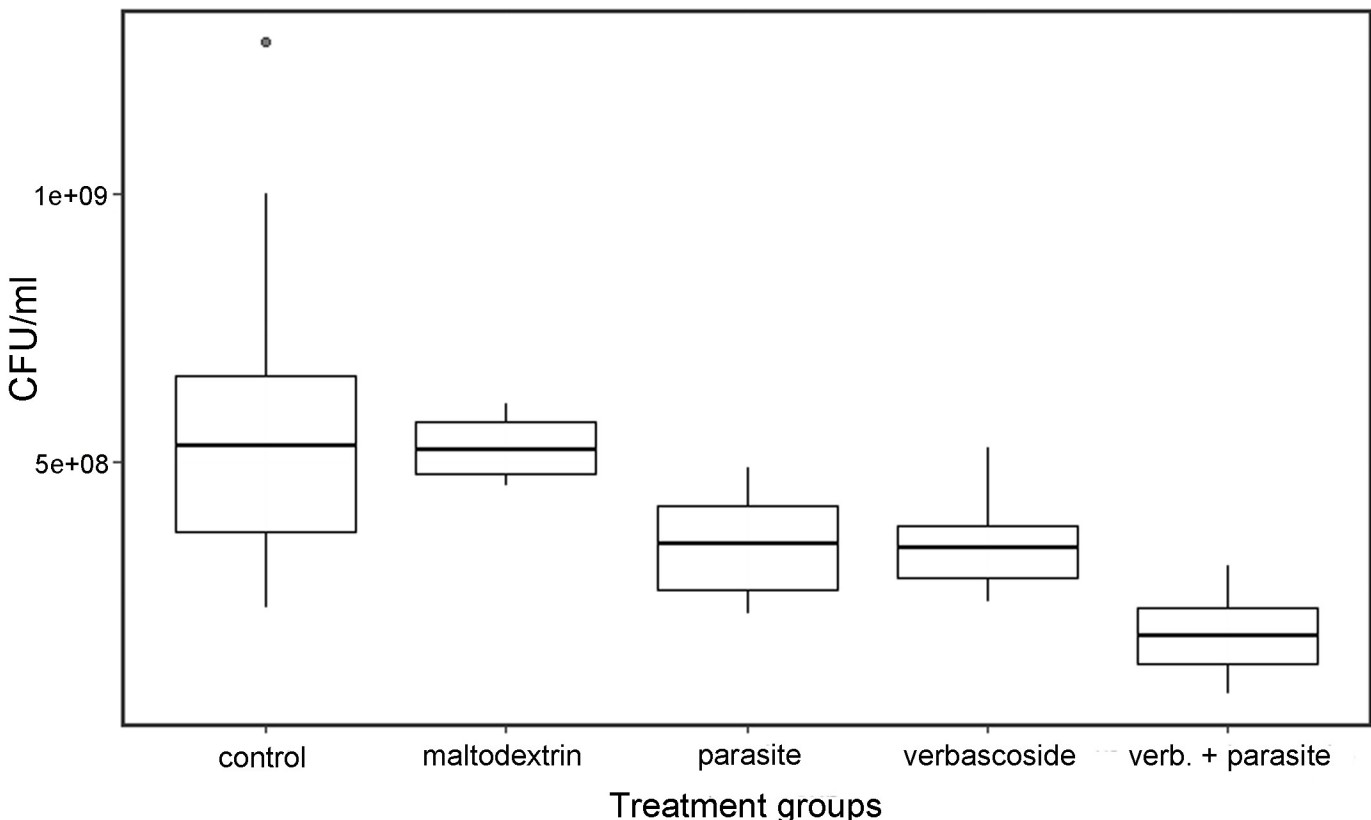

**Fig 5. Comparison of antibacterial activity among control and treatment groups.** Wasps that ingested: water and sugar (Control), water and maltodextrins (Maltodextrin), water, verbascoside and maltodextrins (Verbascoside). Parasitized wasps that ingested: water and sugar (Parasite), water, verbascoside and maltodextrins (Verbascoside+Parasite). Box-plots represent the number of *E. coli* colony-forming units per ml (CFU/ml) for each group.

parasitized wasps on untreated fresh buds. The result clearly suggests that the presence of the parasite *X. vesparum* influences the wasp's attraction for fresh buds, especially in field-collected wasps, probably due to their previous foraging experience on trumpet creepers. Parasite sex did not affect the time that the host spent foraging on fresh buds. However, our observations were performed during an early phase of the foraging season (summer), while later on the

**Table 3. Dunn multiple comparisons among groups following a significant Kruskal-Wallis test.**

| Comparison | Z | p unadj | p adj |
|---|---|---|---|
| maltodextrins*control | -0.500 | 0.616 | 0.685 |
| verbascoside*control | 3.0746 | 0.002 | **0.005**** |
| parasite*control | 2.337 | 0.019 | **0.028*** |
| verbascoside+parasite*control | 6.038 | <0.001 | <**0.001***** |
| verbascoside*maltodextrins | 2.471 | 0.013 | **0.022*** |
| parasite*maltodextrins | 2.165 | 0.030 | **0.038*** |
| verbascoside+parasite*maltodextrins | 4.635 | <0.001 | <**0.001***** |
| parasite*verbascoside | 0.037 | 0.970 | 0.970 |
| verbascoside+parasite*verbascoside | 3.730 | 0.001 | <**0.001***** |
| verbascoside+parasite*parasite | 2.957 | 0.003 | <**0.001***** |

(p-values adjusted with the Benjamini-Hochberg method) ($p < 0.05$*; $p < 0.01$**; $p < 0.001$***).

scenario might change due to different energetic demands of wasps infected by female versus male parasites. In fact, only female-infected wasps can overwinter, due to their large fat bodies, while wasps parasitized by males have scanty lipid storage and will die at the end of summer [19].

The abundance of EFNs on fresh buds of trumpet creeper [23, 24], secreting sucrose, fructose and glucose, has been documented in Bignoniaceae [51, 52], as well as the secretion of verbascoside and further phenylpropanoid glycosides in the genus *Campsis* [53–55]. The peculiarity of trumpet creepers is the high concentration of a single compound, verbascoside, in their fresh floral buds, an unexpected result. Thanks to our in-depth electron microscopy approach we were able to reveal the specialized glandular structure of EFNs supporting the secretion of high amounts of phenolic compounds. Verbascoside is a disaccharide caffeoyl ester and a secondary metabolite, displaying important antibacterial and antioxidant effects. This compound has been detected in more than 200 plant species worldwide [50, 56] and is thought to be produced by the plant as a defence mechanism against parasites and foliage-feeding caterpillars, but its oral toxicity is dose-dependent in insects [57], as well as in other organisms (e.g. vertebrates) [58].

Our study shows that the intake of a low dosage of verbascoside, consistent with a scenario where wasps feeds on buds and are exposed to natural EFNs concentrations, enhances the wasp's immune response. If we combine this observation with the longer time spent on EFNs by parasitized wasps with respect to non-parasitized ones, other major finding of this study, it becomes natural to hypothesize a parasitic adaptive manipulation of host feeding behaviour [9]. This is a subtle strategy operated by several parasites [59–62] like Strepsiptera that are able to consume host resources without seriously compromising host development [63]. In line with the parasite manipulation hypothesis, the consumption of verbascoside might increase the survival of the host, which is the vehicle of the infection, by boosting its immune response against a range of possible pathogens, including bacteria. This can be achieved in several ways, for example by increasing the number of circulating hemocytes, by mounting a stronger antibacterial response or by increasing the expression of the immune genes like *defensin* [33, 64].

An alternative explanation for the attraction of parasitized wasps to trumpet creepers is that they are in poor health conditions and might seek plant secretions simply to feed on the sugary solutions that they contain. However, this explanation seems unlikely, since when the wasp emerges as an adult the parasite stops draining host resources [65]. Moreover, in the laboratory the hourly frequency of foraging for sugar cubes did not differ between parasitized and non-parasitized wasps, regardless of the parasite's sex [66] and field observations showed that parasitized wasps did not visit indiscriminately any flowering bushes but they gathered preferentially on trumpet creepers [17] and other selected plants [15] that are rich in EFNs and possibly in bioactive compounds (i.e., *Morus*, *Vitis*, *Hedera*, *Cynara*, *Populus* spp). As a third possible explanation for the phenomenon we observed in this study, the consumption of EFN secretions containing verbascoside might be interpreted as an example of self-medication against parasites and infections [67, 68]. Nevertheless, this altered feeding behaviour does not appear to have any detrimental effect neither for the strepsipterans nor for non-parasitized wasps, as one would expect in a traditional scenario of therapeutic self-medication [69–71]. On the contrary, the immune response of both non-parasitized and parasitized wasps was enhanced by verbascoside with respect to controls.

In conclusion, we hypothesize that the presence of the parasite and the administration of verbascoside have a synergistic effect against pathogenic infection and may lengthen host life-span: a critical feature for parasite transmission [14, 19, 72]. Moreover, it is possible that *C. radicans*, thanks to EFNs secretions, enhances wasps' foraging behavior and therefore gains defence against insect pests also attracted to trumpet creepers. More studies in the future must

be undertaken to explore the relevant role of wasps in plant defence, a topic that has started to be investigated only recently [73]. If *C. radicans* offers, besides nutrients and shelter, also additive therapeutic extra-floral secretions to wasps in change of anti-herbivore defence, then it would be fascinating to hypothesize the existence of a three-way interaction between wasps, parasites and medical plants.

## Supporting information

**S1 File. Preference bioassays.**
(DOCX)

**S2 File. Verbascoside and chromatographic profiles at different wavelengths.**
(DOCX)

**S1 Video. Parasitized *P. dominula* wasps feeding and defending the buds of *C. radicans*.**
(AVI)

## Acknowledgments

The authors are grateful to Rita Cervo, Stefano Turillazzi and the members of the Florence Group for the Study of Social Wasps, first of all to Irene Pepiciello, for their assistance during this study, both in the field and in the laboratory. The authors also thank two anonymous Reviewers for their helpful comments and suggestions.

## Author Contributions

**Conceptualization:** Laura Beani, Marta Mariotti Lippi, Nadia Mulinacci, Fabio Manfredini, Duccio Cavalieri, Federico Cappa.

**Data curation:** Laura Beani, Nadia Mulinacci, Lorenzo Cecchi, Corrado Tani, Niccolò Meriggi.

**Formal analysis:** Laura Beani, Niccolò Meriggi.

**Funding acquisition:** Laura Beani.

**Investigation:** Laura Beani, Marta Mariotti Lippi, Nadia Mulinacci, Fabio Manfredini, Lorenzo Cecchi, Claudia Giuliani, Corrado Tani, Niccolò Meriggi, Federico Cappa.

**Methodology:** Laura Beani, Marta Mariotti Lippi, Nadia Mulinacci, Fabio Manfredini, Lorenzo Cecchi, Claudia Giuliani, Corrado Tani, Niccolò Meriggi, Federico Cappa.

**Project administration:** Laura Beani.

**Software:** Niccolò Meriggi.

**Supervision:** Laura Beani, Fabio Manfredini, Duccio Cavalieri, Federico Cappa.

**Validation:** Fabio Manfredini, Federico Cappa.

**Writing – original draft:** Laura Beani, Fabio Manfredini, Niccolò Meriggi, Federico Cappa.

**Writing – review & editing:** Laura Beani, Marta Mariotti Lippi, Nadia Mulinacci, Fabio Manfredini, Niccolò Meriggi, Duccio Cavalieri, Federico Cappa.

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
