## [Decision Letter · Decision Letter 0]

21 Oct 2020

PONE-D-20-29450

Altered feeding behavior and immune competence in paper wasps: a case of parasite manipulation?

PLOS ONE

Dear Dr. Cappa,

Thank you for submitting your manuscript to PLOS ONE. After careful consideration, we feel that it has merit but does not fully meet PLOS ONE’s publication criteria as it currently stands. Therefore, we invite you to submit a revised version of the manuscript that addresses the points raised during the review process.

I suggest that you follow comments and suggestions made by the referees. These modifications will improve your manuscript. 

We look forward to receiving your revised manuscript.

Kind regards,

Fabio S. Nascimento

Academic Editor

PLOS ONE

Journal Requirements:

Reviewers' comments:

Reviewer's Responses to Questions

**Comments to the Author**

1. Is the manuscript technically sound, and do the data support the conclusions?

Reviewer #1: Yes

Reviewer #2: Yes

2. Has the statistical analysis been performed appropriately and rigorously? 

Reviewer #1: Yes

Reviewer #2: Yes

3. Have the authors made all data underlying the findings in their manuscript fully available?

Reviewer #1: No

Reviewer #2: Yes

4. Is the manuscript presented in an intelligible fashion and written in standard English?

Reviewer #1: Yes

Reviewer #2: Yes

5. Review Comments to the Author

Reviewer #1: This manuscript investigated aspects of the plant Campis radicans, in which Polistes wasps infected with the parasite Xenus vesparum frequently forage. The manuscript uses and interesting approach combining laboratory bioassays, chemical analysis HPLC-DAD-MS and testing the immunocompetence of one of the compounds found in abundance in extra-floral nectarines. I found this manuscript interesting and rich in details.

I have just a couple of questions and suggestions to improve the manuscript.

At Ln 81, the authors suggest that the investigations motivated by the fact that verbascoside is known for its bioactive properties. Although I see that maybe this was maybe the most important compound found by the authors, I would like to see the identification and quantification of the other big compounds, because it would justify the choice of only test verbascoside (Fig S3).

Another question is, if mostly parasitized wasps are found in the plant, how the compound could increase the immune system by the intake of verbascoside? Although during parasite host manipulation, the parasite Xenus vesparum would manipulate the host to be somehow healthier? Did the authors expect this outcome?

What is the explanation to the fact that small buds have more compounds?

Small comments:

Ln 170: The immune challenge was performed 7 days after the first administration. Why is that?

Ln 172 Delete: (Schmid-Hempel 2005; Charlen and Killian 2015)

Change letters of figures to capital.

About Figure S3, what are those two peaks on the sides of verbascoside? What other compounds were found in the extracts?

At Figure 2, include the sample size.

ln 251: extra “.”

Reviewer #2: The present work is a very interesting and important study on the prevalence of paper wasps (Polistes dominula) parasitized by the strepsipteran Xenos vesparum for extra-floral nectaries which cover the buds of the trumpet creeper Campsis radicans. In this very carefully conducted study, the authors investigated the influence of this plant on wasp behaviour and physiology through a multidisciplinary approach using a) laboratory bioassays, b) describing the structure and ultra-structure the extra-floral nectaries secreting cells, c) analysing extracts from different bud tissues by HPLC-DAD-MS, and d) testing the immune-stimulant properties of verbascoside by measuring bacterial clearance in Polistes wasps, as a proxy for overall immune competence.

The authors could clearly show that the immune competence was significantly increased after administration of verbascosid, even more so the wasps was parasitized by X. vesparum. The authors hypothesise that the parasite manipulates the host's behaviour to feed preferentially on extra-floral nectaries of Campsis radicans, as the bioactive properties of verbascoside are likely to increase host survival and hence parasite fitness.

The manuscript provides sufficient background and introduction to place it into a broader field of knowledge. The text is very well written and is organized into coherent subsections.

I therefore recommend the acceptance of the manuscript. I have added some very minor additions and comments for the authors.

Specific comments

For Campsis radicans (line 56) the author is given, for Xenos vesparum (line 45) and Polistes dominula (line 46) not. To be consistent, I would also indicate the author for Polistes and Xenos.

Line 16: I prefer "primary larvae" instead of "young larvae". This is more specific.

Line 136: If the heading is in italics, genus and species should not be written in italics.

Line 156: C. radicans not italic

Line 251: Double period.

Line 268: C. radicans not italic

Line 457: Campsis in italics

Line 471: Polistes exclamans in italics

Line 515: Radicans, small letters

Line 527: Campsis grandiflora in italics

Figure 3 and corresponding figure caption (lines 593-594): The assignment of the scales is difficult. I would write uniformly, as in B and C, the size above the scale and not in the caption.

Figures 2, 4, 5: The resolution is too low.

6. PLOS authors have the option to publish the peer review history of their article (what does this mean?). If published, this will include your full peer review and any attached files.

Reviewer #1: No

Reviewer #2: No

---

## [Author Response · Author response to Decision Letter 0]

30 Oct 2020

Response to Reviewers

Review Comments to the Author

Reviewer #1: This manuscript investigated aspects of the plant Campis radicans, in which Polistes wasps infected with the parasite Xenus vesparum frequently forage. The manuscript uses and interesting approach combining laboratory bioassays, chemical analysis HPLC-DAD-MS and testing the immunocompetence of one of the compounds found in abundance in extra-floral nectarines. I found this manuscript interesting and rich in details.

I have just a couple of questions and suggestions to improve the manuscript.

At Ln 81, the authors suggest that the investigations motivated by the fact that verbascoside is known for its bioactive properties. Although I see that maybe this was maybe the most important compound found by the authors, I would like to see the identification and quantification of the other big compounds, because it would justify the choice of only test verbascoside (Fig S3).

Response – Following the reviewer suggestion, we clarified this point. The figure reported as Fig. S3 was a 100-times zoom in the range 7-19 minutes of the original chromatographic profile of the extract. To make this more clear for the reader, the original profile is now reported together with the zoom in the figure S3. As it clearly appears with this new figure, there are no big compounds other than verbascoside. However, among the minor peaks, the biggest two (retention time about 12.0 and 12.6 minutes) have been identified as verbascoside related compounds (according to the very similar UV-vis spectra) with mass spectra that indicate a molecular weight of 638 Da and 668 Da. With these data, these compounds can be tentatively identified as ethyl-β-OH-acteoside (668 Da) and eukovoside (638 Da). The amount of the two peaks is below 5% of verbascoside content, consequently, we only quantified verbascoside (see Alipieva et al. 2014 for conflicting reports regarding the designation of verbascoside).

Furthermore, by MS analysis some compounds present in trace amount have been identified as the verboascoside related compounds β-OH-acteoside isomer 1 and β-OH-acteoside isomer 2, and the flavonol rutin. This information was added to supplementary materials (lines 78-90).

Another question is, if mostly parasitized wasps are found in the plant, how the compound could increase the immune system by the intake of verbascoside? Although during parasite host manipulation, the parasite Xenus vesparum would manipulate the host to be somehow healthier? Did the authors expect this outcome? 

Response - In accordance with the reviewer, we hypothesized that the parasite manipulates the feeding behaviour of the host prompting the wasp to search for Campsis bushes to feed on EFNs. Since verbascoside was the most abundant compound found in EFNs’extracts, and given its antioxidant and antibacterial properties, we hypothesized that the ingestion of verbascoside would enhance the host’s ability to clear a bacterial infection. Preference bioassays and Immunecompetence assays, with a marked preference of parasitized wasps for fresh buds and a higher degree of bacterial clearance in wasps treated with verbascoside are consistent with this hypothesis. 

What is the explanation to the fact that small buds have more compounds?

Response - The number of EFNs, covering bud surface, is higher in 7 small buds than in 3 large buds, thus the content of verbascoside expressed as mg of verbascoside per gram of tissue is double in small buds in comparison to large buds, while the concentration is similar if we consider mg of verbascoside per bud (lines 287-292 and Table 2). We hypothesized that the function of elevated concentration of verbascoside (especially in younger, developing buds) is to defend the extrafloral nectaries from consumers (florivores) (lines 290-291). The higher concentration of secondary metabolites in younger/developing plant tissues as reported in:

Achakzai, A. K. K., Achakzai, P., Masood, A., Kayani, S. A., & Tareen, R. B. (2009). Response of plant parts and age on the distribution of secondary metabolites on plants found in Quetta. Pak. J. Bot, 41(5), 2129-2135.

Verma, N., & Shukla, S. (2015). Impact of various factors responsible for fluctuation in plant secondary metabolites. Journal of Applied Research on Medicinal and Aromatic Plants, 2(4), 105-113.

Small comments:

Ln 170: The immune challenge was performed 7 days after the first administration. Why is that?

Response – The immune challenge was performed the 7th day after the first administration because we administered our verbascoside doses on three different times: day 1, 3 and 5, to avoid a high dose in a single feeding event and to simulate a natural foraging, while on day 2, 4 and 6 we left the wasps undisturbed, then on day 7 we performed our immune challenge.

Ln 172 Delete: (Schmid-Hempel 2005; Charlen and Killian 2015)

Response - the correction was made.

Change letters of figures to capital.

Response – the corrections were made.

About Figure S3, what are those two peaks on the sides of verbascoside? What other compounds were found in the extracts?

Response – Please, see the response to the first comment of Reviewer #1 

At Figure 2, include the sample size.

Response - The sample size of wasps tested in Preference biossays is now included in the caption of Fig. 2.

ln 251: extra “.”

Response – deleted.

Reviewer #2: The present work is a very interesting and important study on the prevalence of paper wasps (Polistes dominula) parasitized by the strepsipteran Xenos vesparum for extra-floral nectaries which cover the buds of the trumpet creeper Campsis radicans. In this very carefully conducted study, the authors investigated the influence of this plant on wasp behaviour and physiology through a multidisciplinary approach using a) laboratory bioassays, b) describing the structure and ultra-structure the extra-floral nectaries secreting cells, c) analysing extracts from different bud tissues by HPLC-DAD-MS, and d) testing the immune-stimulant properties of verbascoside by measuring bacterial clearance in Polistes wasps, as a proxy for overall immune competence.

The authors could clearly show that the immune competence was significantly increased after administration of verbascosid, even more so the wasps was parasitized by X. vesparum. The authors hypothesise that the parasite manipulates the host's behaviour to feed preferentially on extra-floral nectaries of Campsis radicans, as the bioactive properties of verbascoside are likely to increase host survival and hence parasite fitness.

The manuscript provides sufficient background and introduction to place it into a broader field of knowledge. The text is very well written and is organized into coherent subsections.

I therefore recommend the acceptance of the manuscript. I have added some very minor additions and comments for the authors.

Specific comments

For Campsis radicans (line 56) the author is given, for Xenos vesparum (line 45) and Polistes dominula (line 46) not. To be consistent, I would also indicate the author for Polistes and Xenos.

Response – the authors are now indicated (lines 45-46).

Line 16: I prefer "primary larvae" instead of "young larvae". This is more specific.

Response - the correction was made.

Line 136: If the heading is in italics, genus and species should not be written in italics.

Response - the correction was made.

Line 156: C. radicans not italic

Response - the correction was made.

Line 136: If the heading is in italics, genus and species should not be written in italics.

Response - the correction was made.

Line 251: Double period.

Response - the correction was made.

Line 268: C. radicans not italic

Response - the correction was made.

Line 457: Campsis in italics

Response - the correction was made.

Line 471: Polistes exclamans in italics

Response - the correction was made.

Line 515: Radicans, small letters

Response - the correction was made.

Line 527: Campsis grandiflora in italics

Response - the correction was made.

Figure 3 and corresponding figure caption (lines 593-594): The assignment of the scales is difficult. I would write uniformly, as in B and C, the size above the scale and not in the caption.

Response - Following the Reviewer’s suggestion, we included the size above the scale and not in the caption.

Figures 2, 4, 5: The resolution is too low 

Response - figures 2, 4 and 5 were modified increasing resolution to 300 pixels/inch

---

## [Editor Report · Decision Letter 1]

4 Nov 2020

Altered feeding behavior and immune competence in paper wasps: a case of parasite manipulation?

PONE-D-20-29450R1

Dear Dr. Cappa,

We’re pleased to inform you that your manuscript has been judged scientifically suitable for publication and will be formally accepted for publication once it meets all outstanding technical requirements.

Kind regards,

Fabio S. Nascimento

Academic Editor

PLOS ONE
---

## [Editor Report · Acceptance letter]

19 Nov 2020

PONE-D-20-29450R1 

Altered feeding behavior and immune competence in paper wasps: a case of parasite manipulation? 

Dear Dr. Beani:

I'm pleased to inform you that your manuscript has been deemed suitable for publication in PLOS ONE. Congratulations! Your manuscript is now with our production department. 

Kind regards, 

on behalf of

Dr. Fabio S. Nascimento 

Academic Editor

PLOS ONE